# Polo-like Kinase 4: A Molecular Culprit in Skin Cancer Pathogenesis

**DOI:** 10.3390/cells14171381

**Published:** 2025-09-04

**Authors:** Tanya Jaiswal, Durdana Muntaqua, Nihal Ahmad

**Affiliations:** 1Department of Dermatology, University of Wisconsin, Madison, WI 53705, USA; tjaiswal@dermatology.wisc.edu (T.J.); durdana@dermatology.wisc.edu (D.M.); 2William S. Middleton Memorial Veterans Hospital, Madison, WI 53705, USA

**Keywords:** polo-like kinase 4, skin cancer, melanoma, Wnt/*β*-catenin, p53, PI3/AKT, NFκB, cGAS-STING, Hippo/YAP, ATR/CHEK1

## Abstract

Skin cancer remains a significant global health challenge, with rising incidence and associated mortality in late-stage and drug-resistant cases. This underscores a continuing need for more effective novel therapeutic options that can be utilized for efficient management of skin cancers. A promising approach involves exploiting novel targets, which are dysregulated in skin cancer, either alone or in combination with existing therapeutics. Among these, polo-like kinases (PLKs), a family of serine/threonine kinases, has emerged as promising candidates due to their essential role in cell cycle and maintaining genomic stability, key hallmarks of cancer. Within this family, polo-like kinase 4 (PLK4) stands out as a structurally distinct member and the master regulator of centriole duplication, ensuring this process occurs only once per cell division. Dysregulation of PLK4 can disrupt genomic integrity, contributing to tumorigenesis, thus making it a promising target for cancer management. Notably, PLK4 is frequently overexpressed in several cancers, including skin cancer, and its precise role in skin cancer is an area of current investigation. Further, several small-molecule PLK4 inhibitors such as centrinone, YLZ-F5, CFI-400945, and RP-1664 have demonstrated efficacy in targeting PLK4. Among these, CFI-400945 has advanced to clinical trials, where it has shown modest anti-cancer activity. In this review, we provide a comprehensive overview of the known functions of PLK4 in skin cancer. Additionally, we discuss potential mechanistic insights into PLK4′s involvement in skin cancer progression by extrapolating evidence from studies in other cancer types including colorectal cancer, thyroid cancer, lymphomas, leukemia, etc., while identifying gaps for future research.

## 1. Introduction

The skin is the largest organ of the human body, serving as a vital barrier against environmental and external threats. However, various genetic, environmental, and internal factors can compromise its integrity, leading to a range of skin conditions. These include inflammatory diseases such as psoriasis, benign growth-like keloids, and more serious conditions such as melanoma and non-melanoma skin cancers. Skin cancer result from an uncontrolled proliferation of neoplastic skin cells and can manifest in different forms, with the most common being melanoma as well as non-melanoma skin cancers (including basal cell carcinoma, BCC and squamous cell carcinoma, SCC). As with other malignancies, skin cancer is characterized by fundamental hallmarks of cancer, including genomic instability [1]. Genomic integrity is a fundamental aspect of cellular homeostasis, and its maintenance is critical for normal functioning and survival [2]. Among the key regulators of genomic integrity, the polo-like kinase (PLK) family, a group of serine/threonine kinases, are pivotal in orchestrating cell cycle progression, mitosis, cytokinesis, and other critical cellular events [3]. The PLK family comprises five members (PLK1–PLK5), all of which share a similar structural framework but differ in their specific cellular functions. Aberrant regulation of PLKs has been implicated in the pathogenesis of several cancers [3], including melanoma and non-melanoma skin cancer, making them candidates for targeted anti-cancer therapy. Recent studies have also highlighted the role of PLK4 in other hyper-proliferative skin conditions such as keloids and psoriasis.

PLKs are characterized by a conserved N-terminal catalytic domain and a C-terminal polo box domain (PBD) (Figure 1a). The PBD is essential for subcellular localization, substrate recognition, and autoregulation of PLKs activity whereas the N-terminal catalytic domain performs the kinase activity [4]. Among these, PLK4 stands out due to its unique structural and functional attributes. It has a triple polo box architecture, which is composed of a cryptic polo box (CPB) domain formed by two non-canonical PBDs beside the conservative PBD [5]. This results in a winged architecture that is crucial for the activation of the kinase, self-regulation, and degradation. The CPB also facilitates homodimerization, enabling PLK4 to interact with a variety of binding partners [5].

Functionally, PLK4 is considered a master regulator of centriole duplication and is intimately involved in cell cycle progression. Through its trans-autophosphorylation and degradation, it limits centriole duplication to once per cell cycle. Its overexpression causes centrosome amplification (CA), genomic instability, and uncontrolled cellular proliferation, which are hallmarks of cancer (Figure 1). PLK4 overexpression has been observed in multiple cancers, including melanoma and non-melanoma skin cancers and is often associated with poor clinical outcomes [6]. The incidence of these cancers has been rising with significant mortality, in the recent past [7]. Although early-stage skin cancers are effectively managed by surgical resection, targeted therapies and immunotherapies, late-stage skin cancers including those with acquired resistance, have dismal prognosis. Therefore, novel strategies and targets are needed for management of late-stage cutaneous malignancies, and PLK4 emerges as a compelling candidate in this context. Although current research linking PLK4 to skin cancer is still in its early stages, preliminary findings are encouraging. They suggest that PLK4 could serve as a viable therapeutic target either as a standalone treatment or as part of a combinatorial approach to combat skin cancer more effectively. Importantly, pharmacological inhibition of PLK4 has demonstrated tumor-suppressive effects in various cancer models. Several small-molecule PLK4 inhibitors, including centrinone, YLZ-F5, CFI-400945, RP-1664 have shown efficacy in targeting PLK4 [6,8,9]. Notably, CFI-400945 has progressed to phase II clinical trials, where it has exhibited modest anti-cancer activity [10].

In this perspective article, we explore the emerging significance of PLK4 in hyperproliferative skin diseases including cancer. We first reflect on the current knowledge of PLK4′s involvement in melanoma, non-melanoma skin cancer, and other dermatologic conditions. Building upon this foundation, we provide perspectives on the potential mechanistic contributions of PLK4 in skin cancer development based on the evidence and insights gained from its role in other malignancies. The subsequent section discusses the current understanding of PLK4′s function in skin-related diseases and skin cancer.

## 2. Polo-like Kinase 4 in Skin Diseases and Skin Cancer

PLK4 being the master regulator of centriole biogenesis plays a crucial role in maintaining genomic integrity, proper chromosomal segregation, and overall ploidy. Aberrant activity of PLK4 causes aneuploidy, which is characterized by abnormal chromosome numbers. It is a hallmark of most tumors and a key indicator of genomic instability. Recent studies leveraging in vitro and in vivo models have shed light on the role of PLK4-mediated centrosome amplification, a frequent cause of aneuploidy, in skin biology and skin carcinogenesis. A recent study from our lab reported that CA is predominantly driven by centriole overduplication, rather than whole-cell doubling or fusion events in melanoma [11]. This distinction was confirmed through immunohistochemical studies using CEP170, a marker of mature centrioles. CEP170 is expected to be found in centrosomes formed via cell doubling but appears less frequently in those generated through centriole overduplication. Additionally, PLK4 was also found to be significantly overexpressed in melanoma tissues and cell lines compared to normal samples [11]. Further, high PLK4 expression in tissue samples from cutaneous melanoma patients was associated with lymph node metastasis, advanced TNM stage, and reduced disease-free survival. Although multivariate analysis did not identify PLK4 as an independent prognostic factor, RNA expression data from the Human Protein Atlas confirmed that high PLK4 levels were associated with shorter overall survival in melanoma patients [12]. While elevated PLK4 levels do not always correlate with centrosome amplification, pharmacological inhibition of PLK4 using centrinone B was shown to suppress proliferation and induced apoptosis in melanoma cells, highlighting PLK4 as a viable therapeutic target for melanoma management [11].

Beyond its role in mitotic fidelity, PLK4 overexpression has been shown to be associated with epidermal hyperplasia and aberrant differentiation in skin. In a transgenic mouse model, PLK4 overexpression induced epidermal thickening manifested as bald patches and loss of melanocyte differentiation, presented as gray hair [13]. The thickened epidermis was indicative of expansion of keratin 5 expressing basal cells into suprabasal layers. These mice also exhibited upregulation of hyperplasia markers like keratin 6 and downregulation of terminal differentiation markers like involucrin, filaggrin, and loricrin. These effects were found to be exacerbated by p53 loss in skin. Importantly, these phenotypes were shown to be associated with loss of primary cilia, suggesting interference with cilia-based signaling pathways and tissue homeostasis [13]. Indeed, this is not unexpected since the centrioles are the foundation upon which cilia are built, and PLK4 is critical in centriole biogenesis. Additional evidence supports that mitotic defects from PLK4 overexpression alone are insufficient to confer a proliferative advantage. A study showed that multi-ciliated cells, spindle orientation errors, and chromosome segregation defects within developing epidermis with PLK4 overexpression, were not sufficient to impart a proliferative advantage if p53 function was intact [14]. These defects triggered p53-mediated cell death and contributed to defective growth and stratification, yet mice remained viable and maintained skin homeostasis without tumorigenesis, underscoring the resilience of epidermal surveillance mechanisms [14]. Moreover, PLK4 overexpression during epidermal development was shown to lead to mitotic spindle orientation defects, multipolar divisions, chromosomal segregation errors, and p53-dependent apoptosis of progenitor cells. Interestingly, while these changes delayed epidermal stratification and compromised the skin barrier, they did not induce spontaneous tumor formation in the surviving mice if PLK4 overexpressing transgene was shutdown postnatally and p53 function was intact. Interestingly, when PLK4 overexpression was coupled with a conditional p53 loss (PLK4OE/p53cKO), some of the observed differentiation defects were partially rescued; however, apoptosis persisted, and aneuploid cells accumulated in the adult epidermis. These mice developed spontaneous skin carcinomas with full penetrance, suggesting that transient centrosome amplification and aneuploidy can drive tumorigenesis in the absence of p53-mediated genomic oversight [15]. Notably, this observed tumorigenic effect appeared to be tissue specific. In contrast to the skin, PLK4 overexpression in mouse liver and fibroblasts, even in the absence of p53, did not consistently result in tumorigenesis or alter thymic lymphoma development that commonly arise in animals lacking p53, underscoring tissue-dependent surveillance mechanisms [16]

PLK4 overexpression has also been linked with damaging effects of ultraviolet (UV) radiation, which is a major risk factor for skin cancer [17]. A bioinformatic-based study of neonatal dermal fibroblasts revealed differential expression of PLK4 depending on recovery time following UV exposure, supporting its involvement in skin carcinogenesis [18]. Nonetheless, it is not clear whether UV exposure triggers PLK4 dysregulation in skin cells or PLK4 overexpression supports UV-mediated skin carcinogenesis. This requires a comprehensive analysis to define the precise relationship between PLK4 and UV exposure in skin cancer, with careful consideration given to extrapolating findings from fibroblasts to keratinocytes or melanocytes, which are directly implicated in skin carcinogenesis.

Recent findings from our lab substantiate the oncogenic role of PLK4 in non-melanoma skin cancer [19]. We observed that PLK4 is significantly overexpressed in BCC and cutaneous SCC cells and tissues. Both genetic knockdown and small molecule inhibition of PLK4 decreased cell viability and proliferation in cSCC and BCC cells and reduced tumor growth in xenograft models while inducing apoptosis and cell cycle arrest and modulating key cancer-related genes in vitro [19]

There is also growing evidence of interplay between inflammatory signaling and PLK4-mediated tumorigenesis [20]. In diseases like psoriasis and cSCC, IL-17-driven inflammation may synergize with PLK4-induced centriole duplication to enhance epidermal proliferation. Genetic models with IL-17 receptor deficiency show reduced epidermal thickening and papilloma formation, while topical PLK4 inhibition alone does not significantly alter disease progression, highlighting a complex interaction between immune and mitotic pathways in skin cancer that needs to be further elucidated [20]. Further, PLK4′s proliferative influence extends beyond malignancy. In keloids, which are benign but aggressively growing fibroproliferative lesions, PLK4 is overexpressed relative to normal skin [21]. Inhibition or knockdown of PLK4 reduced keloid fibroblast proliferation, migration and invasion while inducing apoptosis and G0/G1 phase cell cycle arrest. These findings further support a central role for PLK4 in regulating cell cycle and growth across hyperproliferative skin pathologies [21]

Together, these findings reinforce a model in which PLK4-mediated centrosome amplification and ensuing aneuploidy probably act as conditional tumorigenic drivers, particularly in contexts where tumor suppressor pathways (like p53) are compromised. While centrosome amplification alone is not universally sufficient to drive cancer, it fosters genomic instability that, under permissive conditions such as inflammation or p53 loss and PLK4 overexpression, can lead to tumorigenesis. Figure 1b illustrates the various mechanisms by which elevated PLK4 levels disrupt cellular homeostasis, potentially contributing to cancer development. These insights open avenues for therapeutic intervention targeting PLK4 in both malignant and benign hyperproliferative skin disorders and skin cancer. Selective PLK4 inhibitors such as CFI-495004 and RP-1664 have shown promise in clinical trials, although their effectiveness in skin cancer treatment remains to be fully validated. Future work should focus on delineating the molecular signaling thresholds at which PLK4-mediated centrosome amplification transitions from a tolerated aberration to a cancer-initiating event and exploring combination therapies targeting PLK4 along with other cell cycle regulators and possibly inflammatory mediators. The following section provides a perspective on the potential mechanism of PLK4 in skin cancer, drawing on evidence of its regulation in other cancers and thereby highlighting the need for further research to elucidate specific functions of PLK4 in skin cancer development.

## 3. Prospective Roles of PLK4 in Skin Cancer

As discussed above, PLK4 is overexpressed in skin cancer and is associated with centrosome amplification, chromosomal instability, uncontrolled tumor growth, and poor overall survival [11,12,19]. Studies have demonstrated that PLK4 inhibition either by genetic manipulation or using small molecule inhibitors impedes cancer growth and it is beneficial for cancer management (reviewed in [6]). While the role of PLK4 has been widely explored in some solid tumors [22] its contribution to the pathogenesis of skin cancer remains comparatively understudied. Research highlights the interaction between PLK4 and various signaling pathways in a variety of cancer models. Some of the pathways that explore the association between PLK4 and different cancers such as endometrial cancer, hepatocellular carcinoma (HCC), glioma, lymphoma, neuroblastoma, thyroid cancer, colorectal cancer, etc., include Wnt/β-catenin [23,24], p53 [25], PI3K/AKT [26,27], Hippo/YAP [28] STING/NF-κB [29], ATR/CHEK1 [30] signaling, which are described below. It is notable that most of the above-mentioned pathways are well-established in melanoma and non-melanoma skin cancers, although a direct association of PLK4 with these pathways in skin cancers remains to be elucidated (Figure 2).

### 3.1. Wnt/β-Catenin Signaling

Wnt/β-catenin signaling is crucial for skin development and homeostasis, since this pathway influences the decisions of embryonic and adult stem cells to adopt the various cell lineages of the skin and its appendages [31]. It is required for proper differentiation of melanocytes and their maintenance [31]. Aberrant activation of this Wnt/β-catenin signaling has been reported, both in melanoma as well as non-melanoma skin cancers [32,33]. Interestingly, PLK4-Wnt/β-catenin crosstalk has been reported in other cancers as well. For example, Liao and colleagues have shown that PLK4 knockdown inhibited the Wnt/β-catenin pathway in colorectal cancer cells both in vitro and in vivo, leading to reduced xenografted tumor growth in nude mice [24]. Similarly, centrinone, a PLK4 small molecule inhibitor, has been shown to inactivate Wnt/β-catenin signaling along with decreased cell viability and cell cycle arrest in thyroid cancer cells [34]. Thus, it is speculated that PLK4 inhibition possibly leads to Wnt/β-catenin pathway inactivation, reducing proliferation and tumor growth in skin cancer. Based on this evidence, we propose that PLK4 may reinforce Wnt-driven oncogenic programs in cutaneous malignancies.

### 3.2. p53 Signaling

Mutations in the tumor suppressor gene *p53* are considered an early event in the development of non-melanoma skin cancer. These mutations may result in *p53* inactivation, frequently through missense mutations, a substantial proportion of which bear UV-specific signature mutations, which are characteristic of non-melanoma skin cancer [35]. In case of melanoma, the functional integrity of p53 is commonly impaired, even when the wild-type form is present. This loss of activity is often driven by elevated MDM2 function, an E3 ubiquitin ligase that regulates p53 protein. This results from either upregulated MDM2 expression or mutations that strengthen its interaction with p53 leading to loss of p53 tumors suppressor activity [36]. So far, most studies on skin or skin cancer have discussed the influence of PLK4 dysregulation in relation to disrupted centriole biogenesis and centrosome amplification. A small number of studies have also indicated a connection with p53 loss in skin cancer as addressed above [16]. The tumor suppressor p53 contributes to the transcriptional repression of PLK4 through multiple mechanisms. It can suppress PLK4 by activating the DREAM (dimerization partner, RB-like, E2F, and multi-vulval class B) complex [37], promoting hypermethylation of PLK4 promoter [38], or recruiting histone deacetylases [39]. It was reported that in *p53*-knockout mice, PLK4 overexpression drives uncontrolled cell proliferation and spontaneous tumor formation [15]. In addition, PLK4 may also be implicated in regulating primary cilia function by modulating cilia-related signaling [13]. Overexpression of PLK4 in mice resulted in loss of differentiated melanocytes and epidermal hyperplasia and thickening. These proliferative cells demonstrated centrosome amplification and primary cilia loss, laying the foundation for neoplastic transformation [13].

### 3.3. PI3K/AKT Signaling

PI3K/AKT pathway also plays a role in melanoma initiation and therapy resistance [40]. It is activated in melanoma through *NRAS*-mediated or loss of *PTEN*-mediated signaling [40]. PLK4 and the PI3K/AKT signaling pathway intersect at several critical nodes in skin cancer biology, though a direct relationship between these important pathways is not known. Yet, PI3K/AKT and PLK4 associations have also been studied in glioma and multiple myeloma models [26,27]. Wang et al. showed that AKT1 is a substrate of PLK4, which can phosphorylate AKT1 at three distinct sites S124, T308, and S473 to promote cell proliferation and invasion of glioma cells [26]. Further, Tian et al. demonstrated that PLK4 inhibition via genetic manipulation reduced neuroblastoma metastasis by decreasing PI3K/AKT pathway [41]. It appears that the PLK4 and PI3K/AKT signaling have convergent effects on cell cycle regulation [41]. PI3K/AKT drives metabolic reprogramming and cell cycle progression via mTOR.

Further PLK4 promotes vasculogenic mimicry (VM), a process where aggressive glioblastoma (GBM) cells form vessel-like structures contributing to resistance against anti-angiogenic therapies. PLK4 achieves this by phosphorylating EphA2 at Ser901 and enhancing phosphorylation at Ser897, which activates the EphA2 signaling pathway. This activation stimulates the PI3K-AKT and MAPK pathways, facilitating VM formation and malignant progression. A strong positive correlation was found between PLK4 expression and EphA2 phosphorylation in glioma tissues [42]. PLK4 also influences glucose metabolism in GBM by directly binding to and phosphorylating AKT1 at S124, T308, and S473, thereby activating the PI3K/AKT/mTOR pathway. Phosphorylation at S124 notably enhances S473 phosphorylation, further driving tumorigenesis and metabolic dysregulation. PLK4 expression positively correlates with AKT1 phosphorylation in glioma tissues [26]. Therefore, investigating how PLK4 synergizes with PI3K/AKT/mTOR axis signaling to accelerate skin cancer progression warrants future studies.

### 3.4. Hippo/YAP Signaling

YAP oncoprotein is overexpressed in non-melanoma skin cancer [43,44]. It has been shown to promote BCC and SCC proliferation by regulating the genes associated with extracellular matrix (ECM) remodeling [43] and RAS signaling [44], respectively. Further, YAP is also overexpressed in melanoma and promotes melanoma metastasis [45]. Hippo is a tumor suppressor, which when activated inhibits the YAP oncogene via phosphorylation and cytoplasmic localization. In case of centrosome amplification, cells having extra centrosomes increase the phosphorylation of LATS1/2 (a component of Hippo pathway) and subsequent inhibition of YAP by activating Hippo pathway [46]. It was found that PLK4 inhibitor CFI-400945 possibly works by activating p53 and Hippo/YAP signaling to control cancer growth in diffuse large B-cell lymphoma [28]. This study showed an increase in LATS1 phosphorylation and consistent YAP phosphorylation following PLK4 inhibition. This was shown to be associated with a translocation of YAP from nucleus to cytoplasm, thereby reducing nuclear YAP expression levels and activity suggesting that [28]. Recently, Jia et al. reported a potential tumor suppressing PLK4-associated lncRNA (lncRNA-PLK4) whose expression was downregulated in HCC cells and tissues. The authors found that PARP inhibitor talazoparib-induced lncRNA-PLK4 suppressed proliferation of HCC cells and tumor growth in a xenograft model through YAP inhibition, indicating a tumor-suppressive axis that could be leveraged therapeutically [47]. This indicates a potential interaction between PLK4 and Hippo/YAP signaling. Thus, there is a possibility of an interaction between PLK4 and Hippo/YAP pathway components, or PLK4-mediated constant centrosome amplification works synergistically with the dysregulation of Hippo/YAP signaling allowing oncogenic activity of YAP in skin cancer. This needs to be validated by testing it in different models of melanoma and/or non-melanoma skin cancers.

### 3.5. cGAS-STING Signaling

The cGAS-STING pathway is the double-edge immune signaling mechanism, which on one hand enhances antitumor immunity and promote cell death by stimulating the activity of dendritic cells and T cells. On the other hand, chronic STING activation fosters an inflammatory environment that facilitates tumor growth and metastasis (reviewed in [48]). A study shows that STING and cGAS expressions vary in melanoma cell lines and the genes encoding STING and cGAS are frequently epigenetically silenced through promoter hypermethylation, which makes STING signaling non-functional [49]. The crosstalk between PLK4 and c-GAS-STING pathways was reported in acute myeloid leukemia where PLK4 inhibition using a small molecule inhibitor (CFI-400945) activated antitumor immune response via the c-GAS-STING pathway in *TP53*-mutated leukemia cells. PLK4 inhibitor mediated senescence and upregulated cytokine response was ameliorated when leukemia cells were treated with cGAS or STING inhibitor [50]. Similarly, CFI-400945 was shown to impede proliferation of HCC cells by activating STING/NF-κB axis eliciting antitumor immunity [29]. These studies shed light on the interaction between PLK4 and cGAS-STING pathway, which remains unexplored in the context of skin cancer. Based on these studies, it appears that PLK4 inhibition may cause cancer reduction by promoting antitumor immunity. However, since skin cancers are immunologically hot, it requires in-depth investigation on how modulation of PLK4 will affect STING-mediated immune response and antitumor immunity in skin cancer.

### 3.6. NF-κB Signaling

Chronic inflammation is a known driver of skin cancer, and NF-κB signaling is central to this process [51]. The NF-κB pathway is involved in UV-mediated sunburn reactions resulting in cutaneous swelling, epidermal hyperplasia, and secretion of pro-inflammatory cytokines. This may lead to transcriptional profiles involved in cancer proliferation, invasion, and metastasis [51]. Kim et al. reported that NF-κB pathway mediates skin inflammation and its p65 subunit promotes keratinocyte proliferation and is essential for SCC [52]. In addition, NF-κB was found to be constitutively activated in melanoma and NF-κB inducing kinase (NIK) was shown to modulate melanoma progression through regulation of genes that support cell survival via the β-catenin pathway [53]. Further, it has been shown that NF-κB regulate the expression of cell cycle genes and PLK4 in dermal fibroblasts [54]. Dermal fibroblasts are an important component of tumor microenvironment, which plays a defensive role in the early stages of melanoma [55]. As melanoma progresses, dermal fibroblasts may be reprogrammed by melanoma cells into cancer-associated fibroblasts (CAFs) or melanoma-associated fibroblasts (MAFs), which subsequently promote tumor expansion, invasive behavior, and metastatic potential [56]. Ledoux et al. observed that treating human dermal fibroblasts with NF-κB siRNA resulted in the loss of PLK4 expression. This was confirmed in U2OS and HeLa cells, where NF-κB siRNA reduced PLK4 expression along with decreased cell proliferation [54]. It was also shown that overexpression of NF-κB subunits upregulated the PLK4 promoter activity [54]. This PLK4-NF-κB interaction has also been studied in glioblastoma where PLK4 was found to induce NF-κB transactivation and increased chemoresistance by phosphorylating IKK-epsilon (IKBKE) in glioblastoma cells [57]. It is possible that similar interaction plays a role in skin cancer since both targets have been separately identified to be overexpressed in melanoma and non-melanoma skin cancers [12,19,52,53]. It is possible that the effect of PLK4 on NF-κB in skin cancer may be pro-proliferative, pro-inflammatory, or both. PLK4 and NF-κB appear to engage in a mutual feedback mechanism, modulating each other’s activity. This reciprocal regulatory interaction offers a strong rationale for dual inhibition strategies in skin cancer therapy, although it requires comprehensive mechanistic elucidation.

### 3.7. ATR/CHEK1 Signaling

ATR/CHEK1 signaling is an essential component of DNA damage response (DDR) pathways, which ensure the integrity of DNA and allows DNA repair before it enters the cell cycle. ATR is both an oncogene and a tumor suppressor in the context of cancer [58]. In normal cells, ATR functions as tumor suppressor to maintain genomic integrity. Loss-of-function mutations have been reported in ATR in a subset of melanomas where about 7% of melanoma tumors in the TCGA have mutations in genes that affect the ATR pathway [59]. The researchers irradiated ATR-mutant melanoma cell line with UVB and found defects in activation of ATR downstream targets such as CHEK1 phosphorylation and cell cycle arrest. It was also reported that ATR-mutant tumors generate a pro-inflammatory microenvironment that promotes tumor growth by regulating the immune responses [59]. However, when cancer cells undergo high levels of replication stress, they exploit ATR pathways to sustain their survival thus making ATR pathway activation an oncogenic event. Upregulation of ATR-CHEK1 pathway was reported in a subset of oral squamous cell carcinoma [60]. Dysregulation of PLK4 can intensify this replication stress. In the context of cancer, aberrant PLK4 expression compromises mitotic fidelity and centrosome homeostasis, exacerbates replication-associated DNA damage, and impairs the DDR efficiency, thereby promoting chromosomal instability and tumor progression [61]. Bao et al. observed that the ATR/CHEK1 pathway was involved in the oncogenic functions of PLK4 in HCC and the authors proposed PLK4/ATR/CHEK1 axis as a potential therapeutic target HCC treatment [30]. Since ATR seems to exhibit dual roles in different skin cancers, it is imperative to delineate the exact interaction between PLK4 and ATR/CHEK1 pathways. Additionally, as mentioned above that UV affects the ATR/CHEK1 pathway, it would be interesting to study UV-mediated dysregulation of PLK4/ATR/CHEK1 axis in skin cancer models.

From a unified perspective, PLK4 may drive tumor progression in skin through mechanisms involving metastasis, enhanced cell cycle activity, and disruption of tumor suppressor signaling [62]. Moreover, its speculated interplay with oncogenic pathways such as Wnt/β-catenin, PI3K/AKT, Hippo/YAP, and NF-κB could further potentiate its tumorigenic role in cutaneous malignancies. These pathways have been implicated in PLK4 across multiple cancers; however, their association with PLK4 in skin cancer remains unsubstantiated. While these pathways are separately recognized in skin cancer, skin-specific mechanistic studies are needed to establish their specific link to PLK4 within this context. Given PLK4′s restricted expression in normal tissues and functional indispensability in mitotic regulation, PLK4 represents a promising, yet underexploited, therapeutic target in cutaneous oncology.

## 4. Conclusions

Collectively, PLK4 appears to play a pivotal role in the pathogenesis of skin cancer. While its function has been extensively characterized in various other malignancies, research specifically addressing its molecular signaling in skin cancer remains limited. The potential molecular interactions of PLK4 with pathways such as Wnt/β-catenin, PI3K/AKT, Hippo/YAP, and NF-κB offer promising avenues for future investigation. This may unlock possibilities for the development and evaluation of PLK4-targeted therapies and yield valuable insights for synergistic therapeutic combinations. Hence, detailed preclinical investigations are warranted to establish the upstream as well as downstream mechanisms of PLK4 in skin cancer.

## Figures and Tables

**Figure 1 cells-14-01381-f001:**
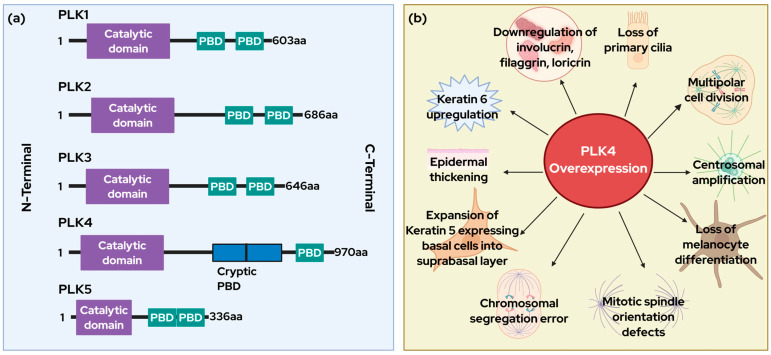
(**a**): Structure of PLKs. This figure illustrates the structures of different members of the PLK family with a conserved N-terminal catalytic domain and two C-terminal polo box domains (PBDs). It also clearly depicts the structural uniqueness of PLK4 with the presence of a cryptic PBD and only a singular conservative PBD. (**b**) PLK4 overexpression as a driver of skin cancer and skin disorders. This figure shows how elevated PLK4 levels disrupt normal cellular processes, leading to mitotic abnormalities, epidermal thickening, impaired melanocyte differentiation, etc. These cellular defects interact with a complex network of known and unknown molecular factors, collectively contributing to skin pathologies and tumorigenesis. Created with BioRender (https://www.biorender.com/, Toronto, ON, Canada).

**Figure 2 cells-14-01381-f002:**
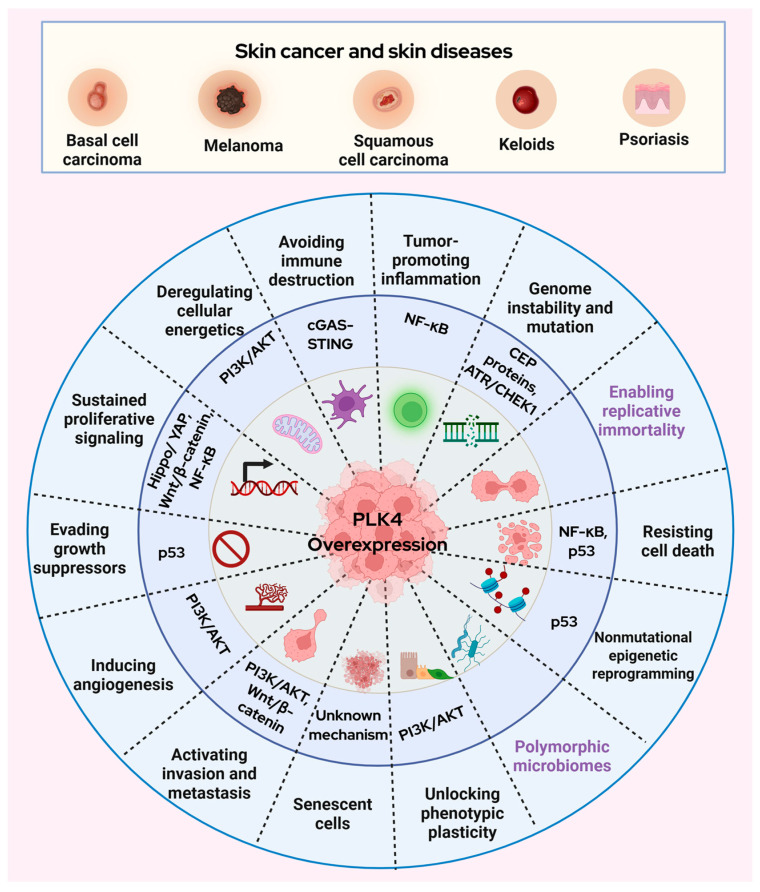
Prospective role of PLK4 in skin cancer. This figure shows the possible interaction between PLK4 overexpression and key signaling pathways in different hallmarks of skin carcinogenesis. PLK4 is well-characterized in other cancer types, where it promotes oncogenic processes through the presented signaling pathways. Although the specific mechanisms by which PLK4 influences these pathways in skin cancer remain unclear. However, given that these same pathways are frequently dysregulated in skin cancer, it is plausible that PLK4 contributes to similar pathological mechanisms. Such dysregulation may promote uncontrolled proliferation, genomic instability, and increased metastatic potential. No possible interactions are known in the hallmarks highlighted in purple. Created with BioRender (https://www.biorender.com/, Toronto, ON, Canada).

## Data Availability

No new data were created or analyzed in this study. Data sharing is not applicable to this article.

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
