# Peer review of "Polo-like Kinase 4: A Molecular Culprit in Skin Cancer Pathogenesis"

_cells, 2025, doi:10.3390/cells14171381_

Round 1
Reviewer 1 Report
Comments and Suggestions for Authors
Jaiswal and co-authors summarized the role of Polo-like Kinase 4 (PLK), a member of the PLK serine/threonine kinases in skin cancer development and the molecular basis. The authors provided a detailed thorough overview of the role of PLK4 in non-melanoma skin cancer and melanoma. The authors’ lab has made significant contribution to this field. The review is well written, and the schematic is well made and illustrates the current understanding of PLK4 in skin cancer and the molecular mechanisms involved.
Minor comments:
- PLK4 interacts with many signaling pathways such as p53, cGAS, and ATR and plays critical roles in SCC, BCC, melanoma, keloids, and psoriasis. It would be helpful to include a table to summarize which pathway interactions have been shown in each skin cancer/diseases.
- P53 dysregulation in SCC/BCC and melanoma is distinct, as majority p53 dysregulation is due to mutations in SCC/BCC, while p53 is silenced by the dysfunction of MDM pathways. It would be helpful to add more discussion on the connection between PLK4 and p53 pathway.
- It would helpful to add more discussion on the overlapping and differential functions of PLK4 from other PLK kinases, including but not limited to PLK1.
Author Response
Comment 1: Jaiswal and co-authors summarized the role of Polo-like Kinase 4 (PLK), a member of the PLK serine/threonine kinases in skin cancer development and the molecular basis. The authors provided a detailed thorough overview of the role of PLK4 in non-melanoma skin cancer and melanoma. The authors’ lab has made significant contribution to this field. The review is well written, and the schematic is well made and illustrates the current understanding of PLK4 in skin cancer and the molecular mechanisms involved.
Response 1: The authors would like to thank the reviewer for their thoughtful reading of our manuscript and providing constructive suggestions. We are submitting the revised manuscript with changes highlighted in yellow.
Comment 2: PLK4 interacts with many signaling pathways such as p53, cGAS, and ATR and plays critical roles in SCC, BCC, melanoma, keloids, and psoriasis. It would be helpful to include a table to summarize which pathway interactions have been shown in each skin cancer/diseases.
Response 2: Thank you for your suggestion. As suggested, we have added a table to summarize the interactions of various pathways with PLK4 in other cancers and the specific roles of those pathways in skin cancers. Since how PLK4 interacts with these pathways in skin cancer is not yet known, we have tried to extrapolate evidence from other cancers.
Comment 3: P53 dysregulation in SCC/BCC and melanoma is distinct, as majority p53 dysregulation is due to mutations in SCC/BCC, while p53 is silenced by the dysfunction of MDM pathways. It would be helpful to add more discussion on the connection between PLK4 and p53 pathway.
Response 3: Thank you for your suggestion. Since, not much is known about the connections between PLK4 and p53 in skin cancers, we have added more discussion on the connection between skin cancers and p53 in lines 245-252. The p53 mutations are considered an early event in non-melanoma skin cancers. The p53 inactivation may occur as a consequence of missense mutations where a significant proportion of these mutations are UV-specific signature mutations in non-melanoma skin cancer. In case of melanoma, the functional integrity of p53 is commonly impaired, even when the wild-type form is present. This loss of activity is often driven by elevated MDM2 function, an E3 ubiquitin ligase that regulates p53 protein. This results from either upregulated MDM2 expression or mutations that strengthen its interaction with p53 leading to loss of p53 tumors suppressor activity.
Comment 4: It would helpful to add more discussion on the overlapping and differential functions of PLK4 from other PLK kinases, including but not limited to PLK1.
Response 4: Thank you for your insightful suggestion. However, discussing the overlapping and differential functions of PLK4 from other PLK kinases is beyond the scope of this article.

Reviewer 2 Report
Comments and Suggestions for Authors
This review provides a comprehensive overview of PLK4 in skin cancer, integrating its structural biology with roles in key signalling pathways such as Wnt/β-catenin, p53, PI3K/AKT, Hippo/YAP, STING/NF-κB, ATR/CHEK1 and DNA damage response. The discussion is thorough and highlights potential therapeutic avenues. Overall, the manuscript is well-structured, timely, and offers valuable mechanistic insights for future research. However, following revisions need to be made before the manuscript is accepted.
- In the final sentence of the abstract add a statement to state the main objective of the review i.e., whether it is to summarize current knowledge, propose hypotheses or identify gaps for future research.
- Emphasize why PLK4 is more promising than other family members in abstract.
- In the abstract mention why this review is timely. Has recent studies, novel inhibitors, or new mechanistic insights have emerged?
- Instead of “ensuring this process occurs only once per cell cycle” in the abstract rephrase "once per cell division" to be precise in cell cycle terminology.
- “extrapolating evidence from other cancer types” briefly hint at which cancer types are most relevant (e.g., breast, colorectal, hepatocellular) in the abstract.
- Lines 158-162. IL-17 influences PLK4 transcriptionally or post-translationally, or indirectly through hyperproliferation. Expand on it.
- Lines 110-115. Clarify if these observations in mouse model have human skin disorder parallels.
- Line 118. Typo error, [[13] should be [13].
- Lines 144–149. Most UV-related studies are in fibroblasts. Note the limitation for extrapolating to keratinocytes/melanocytes.
- Line 175. Figure 1(b) is referenced without explanation. Briefly describe what it depicts.
- line 207: PI3K/AKT [[27,28] remove extra bracket.
- Lines 275-277. Sentence repetition: “… is worth exploring in future studies” appears twice. Mention once.
- Clearly separate whether PLK4’s net effect on Hippo/YAP is activating or suppressive. The mixed mention of tumor-suppressive and oncogenic roles may confuse readers.
- Line 300-303. “double-edged sword” nature of STING is well highlighted, but propose a hypothesis for skin cancer specifically. Would PLK4 inhibition enhance anti-tumor immunity or promote chronic inflammation?
- L320–323: State whether PLK4’s effect on NF-κB in skin cancer context is likely pro-proliferative, pro-inflammatory, or both.
- L328–331. Clearly connect relevance of fibroblast findings to skin tumor stroma or microenvironment.
- L333–339. Indicate whether the PLK4–NF-κB feedback loop could be targeted with dual inhibition for therapeutic benefit.
- L343–345. Briefly connect replication stress and DNA damage response in skin cancers (e.g., UV-induced lesions) to PLK4 dysregulation.
- L351–355. Clarify if UV damage could similarly activate the PLK4/ATR/CHEK1 axis as observed in HCC.
- Explicitly summarize in conclusion the most compelling evidence presented, e.g., Hippo/YAP, cGAS–STING, NF-κB, ATR/CHEK1 involvement.
- 368–370. While the limited research in skin cancer has been mentioned, specify which key gaps, e.g., lack of skin-specific mechanistic studies, absence of in vivo validation etc. are most urgent to address.

Author Response
Comment 1: This review provides a comprehensive overview of PLK4 in skin cancer, integrating its structural biology with roles in key signalling pathways such as Wnt/β-catenin, p53, PI3K/AKT, Hippo/YAP, STING/NF-κB, ATR/CHEK1 and DNA damage response. The discussion is thorough and highlights potential therapeutic avenues. Overall, the manuscript is well-structured, timely, and offers valuable mechanistic insights for future research. However, following revisions need to be made before the manuscript is accepted.
Response 1: The authors would like to thank the reviewer for their thoughtful reading of our manuscript and providing constructive suggestions. We are submitting the revised manuscript with changes highlighted in yellow.
Comment 2: In the final sentence of the abstract add a statement to state the main objective of the review i.e., whether it is to summarize current knowledge, propose hypotheses or identify gaps for future research.
Response 2: Thank you for your suggestion. As suggested, we have stated the main objective of the review in line 31 of the abstract which is to extrapolate evidence of connection between PLK4 and other cancers to skin cancer and to identify gaps for future research.
Comment 3: Emphasize why PLK4 is more promising than other family members in abstract.
Response 3: As suggested, we have emphasized why PLK4 is more promising than other family members in lines 20-22 of the abstract. Dysregulation of PLK4 is known to disrupt genomic integrity and contribute to tumorigenesis. PLK4 dysregulation is also found in various cancers making it a promising target for cancer management.
Comment 4: In the abstract mention why this review is timely. Has recent studies, novel inhibitors, or new mechanistic insights have emerged?
Response 4: As suggested, we have highlighted why this review is timely in lines 24-27 of the abstract. Several novel inhibitors of PLK4 exist, out of which CFI-400945 has advanced to clinical trials making PLK4 a potential target for cancer management.
Comment 5: Instead of “ensuring this process occurs only once per cell cycle” in the abstract rephrase "once per cell division" to be precise in cell cycle terminology.
Response 5: Thank you for pointing this out. We have replaced cell cycle with cell division.
Comment 6: “extrapolating evidence from other cancer types” briefly hint at which cancer types are most relevant (e.g., breast, colorectal, hepatocellular) in the abstract.
Response 6: As suggested, we have highlighted the relevant cancer types i.e., colorectal cancer, thyroid cancer, lymphomas, leukemia etc. That has been discussed in the article. Line 30.
Comment 7: Lines 158-162. IL-17 influences PLK4 transcriptionally or post-translationally, or indirectly through hyperproliferation. Expand on it.
Response 7: Thank you for your suggestion. However, the precise mechanism of how IL17 influences PLK4 in skin cancers is not studied yet and contributes to a knowledge gap in the area.
Comment 8: Lines 110-115. Clarify if these observations in mouse model have human skin disorder parallels.
Response 8: Thank you so much for your insightful suggestion. We agree that extrapolating data from mouse models to human skin has its own shortcomings but discussing that is beyond the scope of this article.
Comment 9: Line 118. Typo error, [[13] should be [13].
Response 9: Thank you for pointing this out. I have removed the extra backet.
Comment 10: Lines 144–149. Most UV-related studies are in fibroblasts. Note the limitation for extrapolating to keratinocytes/melanocytes.
Response 10: Thank you for the suggestion. As suggested, we have noted the limitation of extrapolating UV associated data from fibroblasts to keratinocytes/ melanocytes in lines 152-156. It would require a comprehensive analysis to define the precise relationship between PLK4 overexpression and UV exposure in skin cancer, with careful consideration given to extrapolating findings from fibroblasts to keratinocytes or melanocytes, which are directly implicated in skin carcinogenesis.
Comment 11: Line 175. Figure 1(b) is referenced without explanation. Briefly describe what it depicts.
Response 11: Thank you for pointing this out. We have added an explanation to Figure 1(b) in lines 181-182. Figure 1(b) depicts the different ways by which elevated PLK4 levels disrupt cellular homeostasis that might lead to cancer
Comment 12: line 207: PI3K/AKT [[27,28] remove extra bracket.
Response 12: Thank you for pointing this out. I have removed the extra backet.
Comment 13: Lines 275-277. Sentence repetition: “… is worth exploring in future studies” appears twice. Mention once.
Response 13: Thank you for pointing this out. I have made sure that it appears only once.
Comment 14: Clearly separate whether PLK4’s net effect on Hippo/YAP is activating or suppressive. The mixed mention of tumor-suppressive and oncogenic roles may confuse readers.
Response 14: As suggested, we have clarified PLK4’s net effect on Hippo/YAP in lines 310-312 and 316-320. Hippo is a tumor suppressor, which when activated inhibits YAP via phosphorylation and cytoplasmic localization. YAP acts as an oncogene. It is reported that there is an increase in LATS1 phosphorylation and consistent YAP phosphorylation after treatment with the PLK4 inhibitor. There was also translocation of YAP from nucleus to cytoplasm, which reduced nuclear YAP expression levels and activity. This indicated that PLK4 inhibition can suppress YAP oncogenic activity in B-cell lymphoma model.
Comment 15: Line 300-303. “double-edged sword” nature of STING is well highlighted, but propose a hypothesis for skin cancer specifically. Would PLK4 inhibition enhance anti-tumor immunity or promote chronic inflammation?
Response 15: Since no concrete evidence exists to draw a solid conclusion on whether PLK4 inhibition would enhance anti-tumor immunity or promote chronic inflammation, based on evidence from other cancers, we hypothesize that PLK4 inhibition may cause cancer reduction by promoting antitumor immunity. However, since skin cancers are immunologically hot, it requires in-depth investigation on how modulation of PLK4 will affect STING-mediated immune response and antitumor immunity in skin cancer. Line 331-336
Comment 16: L320–323: State whether PLK4’s effect on NF-κB in skin cancer context is likely pro-proliferative, pro-inflammatory, or both.
Response 16: As suggested, we have included a hypothesis on PLK4’s effect on NF-κB in skin cancer context in lines 361-366 since no concrete evidence exists connecting the two. It is hypothesized that the effect of PLK4 on NF-κB in skin cancer maybe pro-proliferative, pro-inflammatory or both.
Comment 17: L328–331. Clearly connect relevance of fibroblast findings to skin tumor stroma or microenvironment.
Response 17: As suggested, we have given a connection between the relevance of fibroblast findings to skin tumor stroma or microenvironment in lines 348-352. Dermal fibroblasts are an important component of tumor microenvironment, which play a defensive role in the early stages of melanoma. As melanoma progresses, dermal fibroblasts may be reprogrammed by melanoma cells into cancer-associated fibroblasts (CAFs) or melanoma-associated fibroblasts (MAFs), which subsequently promote tumor expansion, invasive behavior, and metastatic potential.
Comment 18: L333–339. Indicate whether the PLK4–NF-κB feedback loop could be targeted with dual inhibition for therapeutic benefit.
Response 18: As suggested, we have discussed whether the PLK4–NF-κB feedback loop could be targeted with dual inhibition for therapeutic benefit in lines 363-366. PLK4 and NF-κB appear to engage in a mutual feedback mechanism, modulating each other’s activity. This reciprocal regulatory effect presents a promising rationale for dual inhibition strategies in skin cancer therapy, though it requires comprehensive mechanistic investigation.
Comment 19: L343–345. Briefly connect replication stress and DNA damage response in skin cancers (e.g., UV-induced lesions) to PLK4 dysregulation.
Response 19: As suggested, we have included a brief description of the connection between replication stress and DNA damage response in skin cancers (e.g., UV-induced lesions) to PLK4 dysregulation in lines 382-385. Dysregulation of PLK4 can intensify this replication stress. In cancer context, aberrant PLK4 expression disrupts mitotic fidelity and centrosome homeostasis, compounding replication-associated DNA damage and impairing DDR efficiency to promote chromosomal instability and tumor progression
Comment 20: L351–355. Clarify if UV damage could similarly activate the PLK4/ATR/CHEK1 axis as observed in HCC.
Response 20: Since no concrete evidence exists to clarify if UV damage could similarly activate the PLK4/ATR/CHEK1 axis as observed in HCC, we have highlighted the research gap in lines 390-392. It would be interesting to study UV-mediated dysregulation of PLK4/ATR/CHEK1 axis in skin cancer models.
Comment 21: Explicitly summarize in conclusion the most compelling evidence presented, e.g., Hippo/YAP, cGAS–STING, NF-κB, ATR/CHEK1 involvement.
Response 21: Since there is no compelling evidence to support the involvedment of a particular pathway with PLK4 in skin carcinogenesis, we suggest further research to address this knowledge gap. Lines 415-419.
Comment 22: 368–370. While the limited research in skin cancer has been mentioned, specify which key gaps, e.g., lack of skin-specific mechanistic studies, absence of in vivo validation etc. are most urgent to address.
Response 22: As suggested, we have included the key gap that is most urgent to address in lines 397-401. The pathways mentioned in this article have been implicated in PLK4 across various cancers; however, their association with PLK4 in skin cancer remains unsubstantiated. While these pathways are separately recognized in skin cancer, skin-specific mechanistic studies are needed to establish their specific link to PLK4 within this context.
